# Electrical Equalization Analysis of PAM-4 Transmission in Short-Reach Optical Systems

**Dana Arie and Gilad Katz ***

Department of Electrical and Electronics Engineering, Holon Institute of Technology, Holon 5810201, Israel; danarie123@gmail.com
* Correspondence: giladka@hit.ac.il

**Abstract:** Inclusive and intensive performance analysis of electrical equalizers in a short-reach optical system using four-level pulse amplitude modulation (PAM-4) is presented in this paper. Two equalizers are used—a feedforward equalizer and decision feedback equalizer using the least mean square algorithm. The sensitivity to cut-off frequency for the transmitter and receiver filters, fiber length and number of equalizers taps in the means of the bit error rate vs. optical input power are shown. The analysis reveals the considerable impact of the filters' bandwidth, particularly in the receiver, on the equalizer performance. These results and their reasons are analyzed and broadly discussed.

**Keywords:** chromatic dispersion; inter-symbol interference; feedforward equalizer; decision feedback equalizer; least mean square

## 1. Introduction

In the last years, we have seen a significant increase in the data transmission rate over optical communication systems, which requires the adaptation of both infrastructure and communication methods. One of these is short-range communication in data centers, which requires especially high rates and capacity. A leading solution seen today is four-level pulse amplitude modulation (PAM-4), which enables working at 100 Gbit/s (and beyond) at a relatively low cost and with less losses [1,2].

Generally, any optic system, and especially at high data-rates, is significantly influenced by the chromatic dispersion (CD), which introduces inter-symbol interference (ISI) along with other ISI contributors in the transmitter and receiver side, mainly due to the limited bandwidth (BW) of their components.

The intensity modulation and direct detection techniques indeed have the advantage of simplicity; however, they can cause critical problems, especially at high rates, such as chirping, signal-dependent noise, and non-linearity of the system. In the presence of ISI, the non-linearity impact is even greater and can stifle the detection of the data.

One of the most common and effective ways to deal with the ISI impairment is the use of electrical equalizers [3,4].

Research on electrical equalizers in optical systems started in the early 1990s [5] and was specifically enhanced around 2000 [3,4,6,7] and until the last few years [8–12]. To our knowledge, this is the first time that the performance analysis of the feedforward equalizer (FFE) and decision feedback equalizer (DFE) in a short-reach optical system has been described, emphasizing the impact of the system critical link parameters: transmitter and receiver low-pass filter (LPF) 3 dB cutoff frequency ($f_c$), fiber length and numbers of the equalizer taps.

Most of the research on electrical equalizers in optical systems neglect the filter BW and the number of taps impact, and those parameters are fixed and not discussed [4,6,9–12]. We note that [3,8] presented a comparison between the variable number of taps but the results and the reason was not discussed broadly; moreover, they referred to long-distance fiber.

The remainder of this paper is divided into four sections. Section 2 presents theoretical details and equations, Section 3 covers the structure and main properties of the simulation, Section 4 shows the results of the simulations, and Section 5 presents the conclusions of this research.

## 2. General Theory

When using a direct modulation, the information is encoded on semiconductor lasers by current modulation. This action, in ideal terms, can be modeled as a square action derived from the square ratio between the electric current and the electric field.

According to Figure 1, we can express the signal along the communication system using Equations (1)–(6) discussed in the following:

$$s(t) = f(\sum_{n=0}^{\infty} I_n h_{TX}(t - nT)), \tag{1}$$

where $I_n$ is a single symbol transmitted at time $n$; $T$ is the symbol period; $h_{TX}$ represents the impulse response of the transmitter, which includes both the modulator driver and the modulator impulse responses; and $f(\cdot)$ represents the modulator electrical current input to optical field output transfer function. The modulator output is followed by a single mode fiber (SMF) with impulse response $c(t)$; thus, the receiver optical input field is presented by Equation (2), where * denotes convolution.

$$r(t) = s(t) * c(t) \tag{2}$$

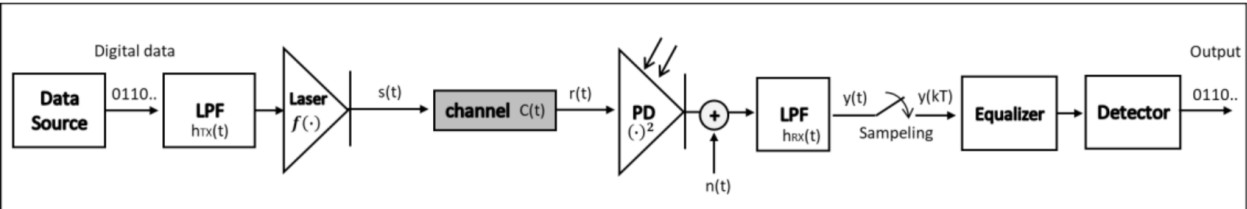

**Figure 1.** Block diagram of an optical communication system.

In the direct detection case, the photodiode square-law operation is followed by a low pass filter (LPF), and the received electrical signal can be expressed as [6]:

$$y(t) = \left(|r(t)|^2 + n(t)\right) * h_{RX}(t), \tag{3}$$

where $n(t)$ is an additive white Gaussian noise (AWGN) and $h_{RX}(t)$ represents the impulse response of the receiver's LPF. Placing Equations (1) and (2) at Equation (3) yields Equation (4):

$$y(t) = \left|f(\sum_{n=0}^{\infty} I_n h_{TX}(t - nT)) * c(t)\right|^2 * h_{RX}(t) + z(t), \tag{4}$$

where $z(t) = n(t) * h_{RX}(t)$.

The sampled signal $y(kT)$ ($k = 0, 1 \ldots$) is fed into the equalizer to mitigate ISI introduced by the dispersion of the SMF, and the transmitter and receiver components' imperfection response. Assuming zero distance fiber, $c(t) = 1$, and $y(kT)$ is [13]:

$$y(kT) = \left|f(\sum_{n=0}^{\infty} I_n h_{TX}(kT - nT))\right|^2 * h_{RX}(kT) + z(kT), \tag{5}$$

In this case, we obtain a linear system. Assuming that the modulator modulates at its linear regime, the optical field output is simply the square-root of the electrical modulator

input current; therefore, $f = \sqrt{(\cdot)}$ [14] and the discrete time equivalent model can be expressed as:

$$y_k = h_{TX,0} I_k h_{RX,k} + \sum_{\substack{n = 0 \\ n \neq k}}^{\infty} I_n h_{TX,k-n} * h_{RX,k} + z_k, \tag{6}$$

The second expression in Equation (6) represents the ISI among a number of adjacent symbols [13], the first expression represents the desirable symbol, compatible to sample $k$, and $z_k$ is the $k$th sample of $z(t)$.

The equalizer's purpose is to compensate for the ISI by finding the reverse channel response while considering the noise power. The ISI is in the model in Equation (6), while $c(t) = 1$ is linear. However, in practice, we do have a fiber, so, definitively, $c(t) \neq 1$, and the ISI, as the system itself, is "non-linear". In a case of non-linear ISI, the use of the FFE or DFE is suboptimal compared to the maximum-likelihood sequence-estimation (MLSE) equalizer.

In this research, we used the mean square error (MSE) criterion through the least mean square (LMS) algorithm for the coefficients' adaptation and optimization. As shown in [13], the filtered gradient LMS algorithm is given by Equation (7).

$$\vec{c}_{k+1} = \vec{c}_k + \Delta \varepsilon_k \vec{V}_k^*, \tag{7}$$

$\vec{c}_k$ is the equalizer's coefficients vector for the $k$th iteration, $\Delta$ is the convergence factor chosen as small enough to ensure convergence of the iterative procedure, $\varepsilon_k = I_k - \hat{I}_k$ is the error signal at the $k$th iteration, and $\hat{I}_k$ is the equalizer output at the same iteration, while $\vec{V}_k$ is the equalizer signal samples input that make up the estimate $\hat{I}_k$. $\vec{V}_k^*$ is the complex conjugate of $\vec{V}_k$.

In our case, the channel is an optical SMF. We can model the SMF as a filter with a fractional response [15] as shown in Equation (8), or in time domain in Equation (9). These equations describe the CD impact on the signal that leads to pulses' expansion and so causes ISI.

$$G(z, \omega) = \exp\left(-j \frac{D\lambda^2}{4\pi c} \omega^2 z\right), \tag{8}$$

$$g(z, t) = \sqrt{\frac{c}{jD\lambda^2 z}} \exp\left(j \frac{\pi c}{D\lambda^2 z} t^2\right), \tag{9}$$

where $z$ is the distance of propagation, $t$ is time variable in a frame moving with the pulse, $\omega$ is the frequency variable, $\lambda$ is the wavelength, $c$ is the speed of light, and $D$ is the dispersion coefficient of the fiber.

## 3. Methods—The Computer Simulation

As mentioned in Section 1, our analytic tool is a computer simulation. This section gives a condensed review on the simulation, its structure and logic, and some defined properties and parameters.

The code models an optical communication system as described in Figure 1. The model runs over four changing parameters' dimensions as described in Figure 2, while the first two dimensions—$f_c$ and the optical input power (OIP), are manifested in the transmission as a 2-dim matrix, and the remaining dimensions are applied (with a "for" loop) on the cannel and equalizer sections. All the parameters are defined in vectors that contained the desirable values for examination.

In order to apply the channel as a filter, we had to define a finite response that suits the infinite response in Equations (8) and (9). According to Savory [15], we can set the

length of the response—$N$, depending on the fiber length, the dispersion parameter, and the wavelength, as given in Equation (10).

$$N = 2 \times \left\lfloor \frac{|D|\lambda^2 L}{2cT^2} \right\rfloor + 1, \tag{10}$$

$L$—the fiber length.

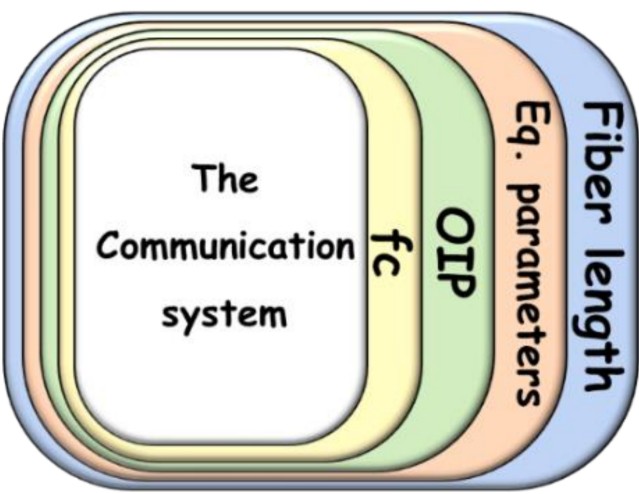

**Figure 2.** The organization of the code running.

At the end of running, all the bit error rate (BER) results are arranged in 4-dim array. This array is being re-arranged into 2-dim matrixes sets that correspond to optical input power (OIP)-defined values. Actually, each combination of the three other parameters has its own matrix. The results are visualized in BER curves as shown in Section 3, Figure 3b. Each set of equalizer taps, where $N_1$ refer to the feedforward part, and $N_2$ refer to the feedback part, owns a different curve pattern, and each LPF 3 dB cutoff ($f_c$) is represented in a separate figure.

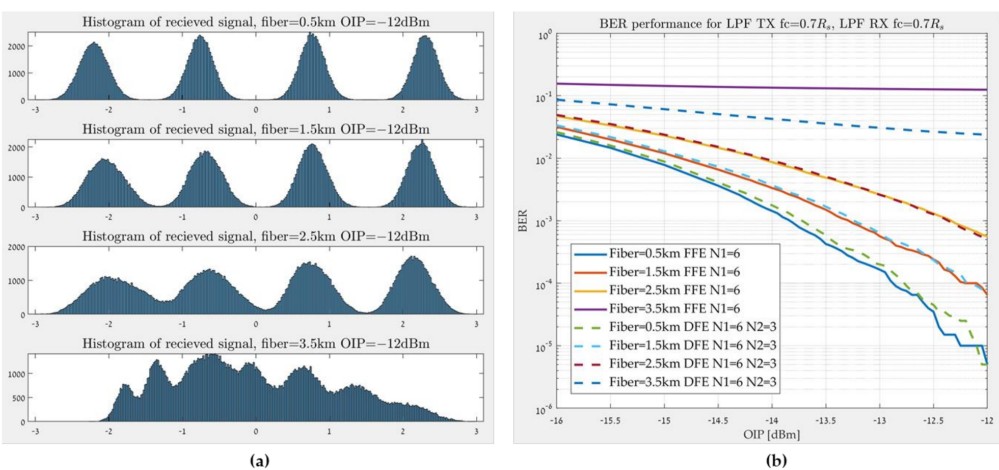

**Figure 3.** (**a**) histograms of the received signal, before the equalizer with different fiber length = 0.5, 2.5, 3.5 km. $f_c = 0.75\ R_s$ Hz, OIP = $-10$ dBm. (**b**) BER curve for three different fiber length = 0.5, 1.5, 2.5, 3.5 km, and two equalizers FFE with $N_1 = 6$, DFE with $N_1 = 6$, $N_2 = 3$. RX $f_c = 0.7R_s$, TX $f_c = 0.55R_s$.

We defined the constant parameters according to [2] and used it to compare the results and for quality check. Table 1 lists those parameters.

**Table 1.** The constant parameters that were defined in the research.

| Parameter | Value |
|---|---|
| Symbol rate $R_s$ | 53 Gbaud |
| Samples per symbol—SPS | 8 |
| Wavelength $\lambda$ | 1550 nm |
| Dispersion parameter $D$ | $17 \times 10^{-3} \frac{s}{m \cdot km}$ |
| Input referred noise—IRN | $20 \times 10^{-12} \frac{A}{\sqrt{Hz}}$ |
| Photodiode responsivity $R_{esp}$ | $0.95 \frac{A}{W}$ |
| Filter order (Bessel) | 4 |
| Convergence parameter $\mu$ | 0.005 |

## 4. Results and Discussion

The first and expected result that was consistent throughout all simulations is the increase in the BER as the fiber was longer. We examined four different fiber lengths, as described in Figure 3, that correspond to different values of the CD parameter $D \cdot L \left[ \frac{ps}{nm} \right]$. A different wavelength will lead to a different $D$ [16], which corresponds to a slight change in fiber length. Figure 3a shows that in 3.5 km (lowest subplot), the histogram of the signal before the equalizer ($y(kT)$) is completely distorted and does not correspond to the original voltage levels of the symbols. According to Figure 3b, even with the DFE, the BER is around 0.02.

### 4.1. The Optimum Equalizer

The first step of the research was to find the ideal numbers of taps. During the research, we tried to investigate its impact on the equalizer's performances. We did not find any great improvement with the increase in the number of taps. In fact, we witnessed a degradation in some of the cases. Figure 4a shows some of the results for the FFE. Clearly, the penalty is higher as the fiber is longer. The independency of the penalty on the filter BW ($f_c$) will be discussed in Sections 4.2 and 4.3.

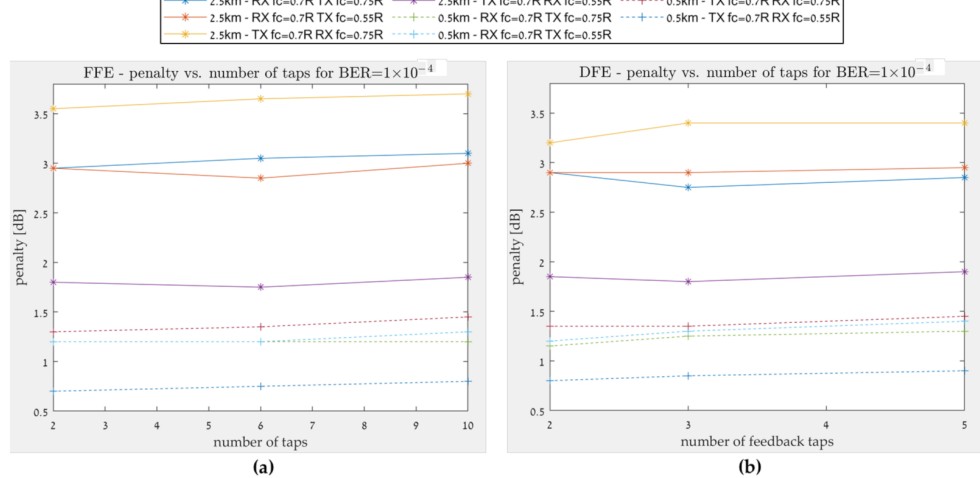

**Figure 4.** Impact of number of taps on the penalty for desirable BER = $1 \times 10^{-4}$, where 0 dB penalty corresponding to OIP of $-14$ dBm. (**a**) FFE with $N_1 = 2, 6, 10$ (**b**) DFE with $N_1 = 6$ and $N_2 = 2, 3, 5$. Fiber length 0.5, 2.5 km, variable combination of filter $f_c$ as mentioned in the legend.

For 2.5 km, the results are inconclusive, so we chose to carry out the rest of the research with six taps at the feedforward part.

A compatible test was performed on the DFE, so six taps were set in the feedforward part. Figure 4b shows these results.

The increase in the feedback taps, as well as in the feedforward part, did not constantly increase the performance, and for the five taps, there is a degradation around 0.1 dBm. In

some of the cases, at 2.5 km fiber length, we can see an improvement with the feedback addition; however, it was about 0.1–0.3 dBm and was inconsistent. For 2.5 km, we chose to carry out the rest of the research with three taps at the feedback part.

### 4.2. Part 1—Conventional Model

The first part relates to the $f_c$ impact and details an analysis with the FFE and DFE on a conventional system model as described above and in Figure 1.

We specified the $f_c$ of one of the sides, transmitter (TX) or receiver (RX), to a normal value of $f_c = 0.7R_s$, where $R_s$ is the symbol rate (*baud*), and changed the other side's $f_c$. This test was meant to separate the ISI that was created in the TX from the one created in the RX because it is critical to discern the difference between them. Due to the square operation of the photodiode, the ISI that is caused by the TX filter and the fiber CD can be considered as "non-linear" ISI. However, at the output of the RX filter, until the equalizer input, we can consider the system as linear, and so is the ISI. We expected that in a case of narrow filter in the TX, the "non-linear" ISI would stifle the equalizer compensation in comparison to a case with an ideal filter width.

Figure 5 shows the opposite.

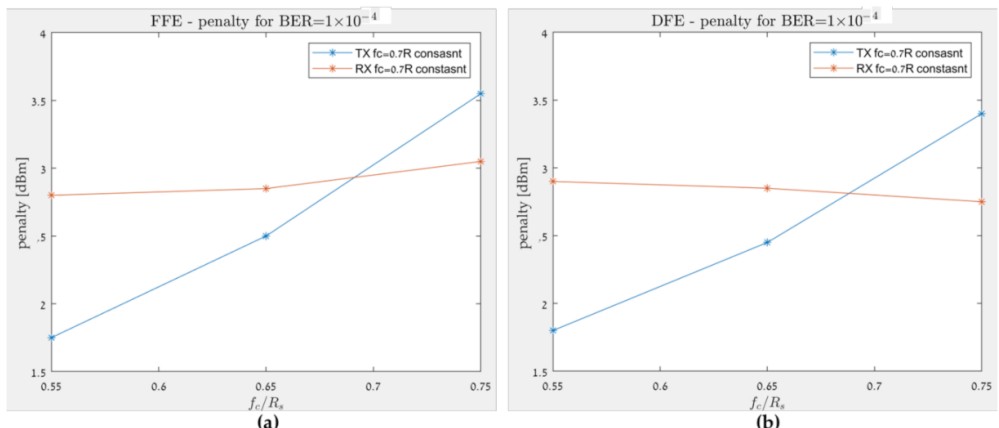

**Figure 5.** Penalty graph for desirable BER = $1 \times 10^{-4}$, where 0 dBm OIP penalty corresponding to $-14$ dBm. (**a**) FFE with $N_1 = 6$ (**b**) DFE with $N_1 = 6$ and $N_2 = 3$.

The orange curve shows that changing the TX filter has a negligible impact on the performances, compared to the RX changes, in the blue curve, that had a more considerable impact. In practice, we obtained better results with the narrow RX filter, and there is a 1.6–1.8 dB penalty for using the wider filter in the receiver. At this point, we assumed that the noise filtering in the RX side, which contributes to the filter narrowing, contributes to the improvement of the results.

Another relevant explanation for the minor impact of the TX's filter changing is the dual influence of the BW. On the one hand, a narrower filter produces more expansion in the time domain and creates ISI. On the other hand, at the narrow filter output, we obtain the signal with lower frequency ingredients; therefore, the CD at the fiber has less impact on the signal expansion, so we obtain less ISI.

### 4.3. Part 2—Isolated Noise Model

To confirm the explanation in the previous subsection about the effect of the noise, we performed another test and moved the noise source to the filter's output and isolated it. As shown in Figure 6, we still obtained better results with the narrow RX filter; however, its advantage decreased and the penalty for using a wider filter dropped to 0.65 dB and 1.2 dB to DFE and FFE, respectively (compared to 1.6 dB and 1.8 dB).

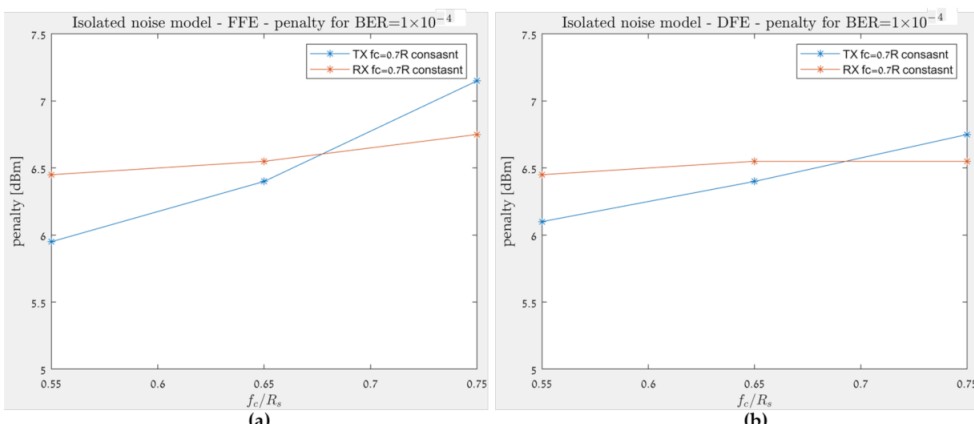

**Figure 6.** Isolated noise model—penalty graph for desirable BER = $1 \times 10^{-4}$, where 0 dBm OIP penalty corresponding to $-14$ dBm. (**a**) FFE with $N_1 = 6$ (**b**) DFE with 6 $N_1 = 6$ and $N_2 = 3$.

On top of the negligible impact of the TX filter, the isolation of the noise resulted in an insignificant impact of the RX filer changing as well. From this part, we can understand that the main cause of the equalizer's reduced performances is the noise, and not only the ISI or the non-linear ISI.

Compared to the results in the previous subsection, the required penalty is doubled and even tripled in some of cases; this emphasizes the filter's critical part as a noise reducer.

## 5. Conclusions

The main finding from the results is the equalizer's sensitivity to the presence of noise. The equalizer should handle the ISI and the channel response, while considering the noise and making a relative compensation when using the MSE criterion. In practice, as evident from the last two sections, we saw that the noise has a dominant impact on the equalizer performance.

Both FFE and DFE perform well with the non-linear ISI caused by the direct detection. Increasing the ISI from the TX filter or the CD did not degrade the performances.

### 5.1. FFE

The addition of the taps in the feedforward part did not affect the performances consistently. The advantage of the addition of taps manifested notably in the high fiber length of 2.5 km.

Increasing the number of taps enlarges the observation range in the time domain, and the resolution in the frequency domain; however, it did not always yield a performance improvement. In the case that a certain number of taps is optimum, any addition on top of that number will not induce any improvement and may even degrade the results because of unnecessary manipulation on the signal.

The FFE in this research is linear, so the taps are spaced at $T_s$. This tap spacing is reciprocal, in frequency domain, to the symbol rate, and it is not compatible with the Nyquist criterion of sampling, which means we would like to demand sampling space $T' = \frac{T_s}{2}$. In general, this approach leads to an equalizer performance that is very sensitive to the choice of sampling time; this may be the reason for the noise sensitivity and the lack of taps increasing the contribution.

In order to eliminate the equalizer sensitivity, it is possible to use a fractionally spaced equalizer (FSE). In the FSE, the taps are spaced at $T' = \frac{M \cdot T_s}{N}$, where $N > M$. The most used space is one that is compatible with the Nyquist criterion $T' = \frac{T_s}{2}$ [13]. Theoretically, the decrease in time space enables high-frequency equalization and can contribute to neutralizing the noise. In practice, because we are discussing a very high-rate communication, it is impossible or very expensive to implement this kind of equalizer.

## 5.2. DFE

The feedback growing does not have any advantage in most of the cases. As well as in the FFE, the advantage manifested notably in the fiber length of 2.5 km. In other conditions, the addition led to a small performance deterioration, or it did not affect the performance at all.

The feedback addition is supposed to improve the equalizer's performance by fully neutralizing the ISI caused by previous symbols. However, the disadvantage of this neutralizing is energy and data loss from previous samples. As the equalizer eliminates the ISI from the symbols that were decided in the detector, we profit from a better compensation; on the other hand, we lose details about other previous symbols. These data are critical to the detection reliability and noise reduction. In fact, the feedback increases the equalizer's sensitivity to the noise, stemming from the spacing of the taps, as explained in the subsection above.

Since we saw that the FFE copes reasonably well without the feedback, as we discussed short-reach fibers, the feedback addition has no contribution to the performance, and it can actually only degrade it.

**Author Contributions:** Conceptualization, G.K.; methodology, G.K. and D.A.; validation, G.K.; investigation, D.A.; writing—original draft preparation, D.A.; writing—review and editing, G.K.; All authors have read and agreed to the published version of the manuscript.

**Funding:** This research received no external funding.

**Institutional Review Board Statement:** Not applicable.

**Informed Consent Statement:** Not applicable.

**Data Availability Statement:** Not applicable.

**Conflicts of Interest:** The authors declare no conflict of interest.

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
