# Peer review of "Electrical Equalization Analysis of PAM-4 Transmission in Short-Reach Optical Systems"

_applsci, doi:10.3390/app12042255_

Round 1
Reviewer 1 Report
1) The meaning of the acronyms(CD, ISI, etc) used should appear in the keywords since they have not been used and therefore explained in the abstract.
2) The LPF acronym is not explained before used (Line 44 in the text)
3) Section 2 is accurate and well presented, but it would be desirable for
a better understanding if the equations used were referenced. In the text
lines 78, 82, 88, to cite some of them where the expresion "eq. " appear.
4) In the same way, the expression "OIP" is used (Line 107) without previous explanation (Line 118).
5) The Results and Discussion section would appear as number 4 and the Conclusions as number 5.
6) Both final parts (Discussion and Conclusions) are consistent and offer a good explanation of the results obtained.
7) The number of references could be somewhat higher for a better understanding
Reviewer 2 Report
A very interesting article on the topic of fiber optic transmission optimization. The presented article shows how the length of the transmission path influences the signal distortion and how it can be reduced using various methods. Reading the article, I missed a broader view of the problem described, i.e. - how will the results change for longer distances of the fiber optic line? - how does the characteristic of the signal coming from the laser diode change as a function of the frequency of the laser current?
